# Design and 22-step synthesis of highly potent D-ring modified and linker-equipped analogs of spongistatin 1

Linda M. Suen[1], Makeda A. Tekle-Smith[1], Kevin S. Williamson[1], Joshua R. Infantine[1], Samuel K. Reznik[1], Paul S. Tanis[1], Tyler D. Casselman[1], Dan L. Sackett[2] & James L. Leighton [1]

Spongistatin 1 is among the most potent anti-proliferative agents ever discovered rendering it an attractive candidate for development as a payload for antibody–drug conjugates and other targeted delivery approaches. Unfortunately, it is unavailable from natural sources and its size and complex stereostructure render chemical synthesis highly time- and resource-intensive. As a result, the design and synthesis of more acid-stable and linker functional group-equipped analogs that retain the low picomolar potency of the parent natural product requires more efficient and step-economical synthetic access. Using uniquely enabling direct complex fragment coupling crotyl- and alkallylsilylation reactions, we report a 22-step synthesis of a rationally designed D-ring modified analog of spongistatin 1 that is characterized by $GI_{50}$ values in the low picomolar range, and a proof-of-concept result that the C (15) acetate may be replaced with linker functional group-bearing esters with only minimal reductions in potency.

[1] Department of Chemistry, Columbia University, New York, NY 10027, USA. [2] Eunice Kennedy Shriver National Institute of Child Health and Human Development, National Institutes of Health, Bethesda, MD 20892, USA. Correspondence and requests for materials should be addressed to D.L.S. (email: sackettd@mail.nih.gov) or to J.L.L. (email: jll43@columbia.edu)

Among the large group of natural products known to express their anti-cancer activity by disrupting microtubule dynamics, spongistatin 1 (refs. [1–3]) is particularly notable for its extraordinary potency, with an average $GI_{50}$ value against the NCI panel of 60 human cancer cell lines of 0.12 nM[1,4]. While the clinical utility of this class of compounds is well established, such ultra-potency is often associated with dose-limiting toxicities that render them unsuitable for use as standalone chemotherapeutics. Conversely, antibody–drug conjugates (ADCs) target tumor-specific antigens, thereby selectively delivering their cytotoxic payload to the cancer cells and effectively increasing its therapeutic index and minimizing off-target toxicities[5–8], and because so little drug material is delivered to the cancer cells, it is not only possible but advantageous to employ ultra-potent cytotoxins that might otherwise be too toxic for use as cancer drugs as ADC payloads. Thus, spongistatin 1, which has been shown to exhibit significant in vivo antitumor activity in an orthotopic pancreatic cancer mouse model[9] and in a melanoma xenograft model[10], is an attractive candidate for development and evaluation as the drug component of ADCs and other selective cancer cell-targeting conjugates.

The development of a natural product such as spongistatin 1 as a payload for selective cancer cell-targeting conjugates entails the installation of a linker functional group that can be used in efficient, mild, and selective conjugation reactions in a region of the molecule that is not critical for activity such that neither the installation of the linker functional group nor its chemical modification in conjugation reactions result in any significant loss of potency. Additionally, since conjugate payloads must survive prolonged exposure to the low pH environments of the endosome and lysozyme, any acid-sensitive groups are likely to be major pharmacological liabilities and must be chemically modified, again in a way that results in no significant loss of potency. This kind of precision modification of natural products[11–15] to develop them as medicinal agents with enhanced pharmacological properties can be addressed only by total chemical synthesis, and though several ground breaking syntheses of spongistatin 1 and 2 have been reported[16–35], spongistatin's size and structural and stereochemical complexity necessitate unusually long synthetic sequences that could render such ADC payload development efforts prohibitively laborious and time- and resource-intensive. We therefore set out to devise new strategies and methods that would result in synthetic access to the spongistatins with greatly increased efficiency and step economy, and then to leverage that synthetic efficiency to address the medicinal chemical development of spongistatin 1 as an ADC payload. Here, we report the development of direct complex fragment coupling crotylation and alkallylation reactions, their application to the development of an efficient and step-economical synthesis of the spongistatins, and the use of that synthesis in the design and validation of analogs that incorporate a pharmacologically innocent linker functional group and a redesigned acid-stable CD spiroketal that is both easier to synthesize efficiently and equipotent with the natural product.

## Results

**Analog design**. Ideally, the identification of a suitable site for the installation of a linker functional group as well as the structural redesign of any acid-sensitive groups with no loss of potency would be guided by structural biological information regarding the interaction of spongistatin 1 with its receptor, but no such information is available. We thus turned to Smith's report that a "diminutive congener" (2) in which the entire C(12)–C(28) region was replaced with a simple linear alkanoate strap retained some significant potency (Fig. 1a)[36]. This suggested to us that the deleted portion of the molecule primarily plays an indirect scaffolding role to favor the biologically active conformation and

reduce the entropic cost of binding, and in turn that relatively minor structural modifications may be made in this region with no significant loss of potency. That the CD spiroketal—whose primary role according to this hypothesis is to properly orient the **R** and **R′** groups—is located in this part of the molecule was of particular interest because it occurs in its thermodynamically less stable singly anomeric form and is susceptible to acid-catalyzed isomerization to the thermodynamically more stable doubly anomeric form with an attendant major change in the relative orientation of the **R** and **R′** groups (Fig. 1b). Indeed, all attempts to synthesize it directly by way of an acid-catalyzed spiroketalization reaction of dihydroxy ketones such as 3 lead to the doubly anomeric form as the major product necessitating an inefficient and laborious separation/equilibration/separation sequence (Fig. 1b). We hypothesized that a redesigned CD* spiroketal in which the D-ring oxygen atom has been relocated to the anomeric position and the C(25) hydroxyl group has been deleted would be (1) readily accessible by efficient and selective spiroketalization of dihydroxyketone 4; (2) immune to acid-catalyzed isomerization as it is in its thermodynamically most stable doubly anomeric form; and (3) isostructural with the natural CD spiroketal in terms of the relative orientation of the **R** and **R′** groups, resulting in no significant reduction in potency. In seeking to identify a group that may be amenable to modification to incorporate a linker functional group with no deleterious impact on potency, we noted that the C(15)-OAc group is also located within the deleted region (Fig. 1c). Our focus on this group as a pharmacologically innocent linker group installation site was further supported by the fact that spongistatins 4 and 6, which differ from spongistatins 1 and 2 only in having a C(15)-OH group, are ~equipotent with spongistatins 1 and 2, respectively, while spongistatin 9, in which the C(15)-oxygen is cyclized onto the C(13) alkene to form a tetrahydrofuran ring is also ~equipotent with spongistatin 1 (ref. [34]). In stark contrast, the C(5)-OAc appears to be important for activity in that spongistatin 3 with a free alcohol at C(5) is about an order of magnitude less potent than spongistatin 1. Together, these data suggest that the C(15)-acetate is not critical for activity, and our aims thus evolved as (1) the development of more step-economical methods and strategies for the synthesis of the spongistatins; (2) synthesis and evaluation of the D-ring-redesigned analog 5a (R=Me); and (3) synthesis and evaluation of azide-equipped ester analogs 5 (R=linker-$N_3$).

**Retrosynthetic analysis**. Retrosynthetic disconnection by way of macrolactonization and Wittig reactions[19] leads to two major fragments of approximately equal complexity, the ABCD* aldehyde 6, and the EF phosphonium salt 7 (Fig. 2). Within both of these fragments may be found homoallylic alcohols (highlighted in magenta) that could in principle be the products of a syn-crotylation reaction of crotylmetal(loid) 8 with aldehyde 9a or 9b, and an alkallylation reaction of allylmetal(loid) 10 with aldehyde 11, respectively. While we saw in these hypothetical transformations great potential to streamline the synthesis, such direct (i.e. generation and use, without isolation, of the active crotyl-/allylmetal species) and highly stereocontrolled complex fragment coupling reactions wherein the crotyl or allyl donor is stereochemically and functionally complex were unknown. The critical methodological challenge was thus the generation of versions of 8 and 10 possessed of the necessary reactivity to couple efficiently and highly stereoselectively with aldehydes 9 and 11 from simple and readily available precursors and in a way that does not require their isolation. We were optimistic that our diamine- and diaminophenol-activated crotyl- and allylsilanes[37,38] would, uniquely, be well-suited for the proposed reactions due both to

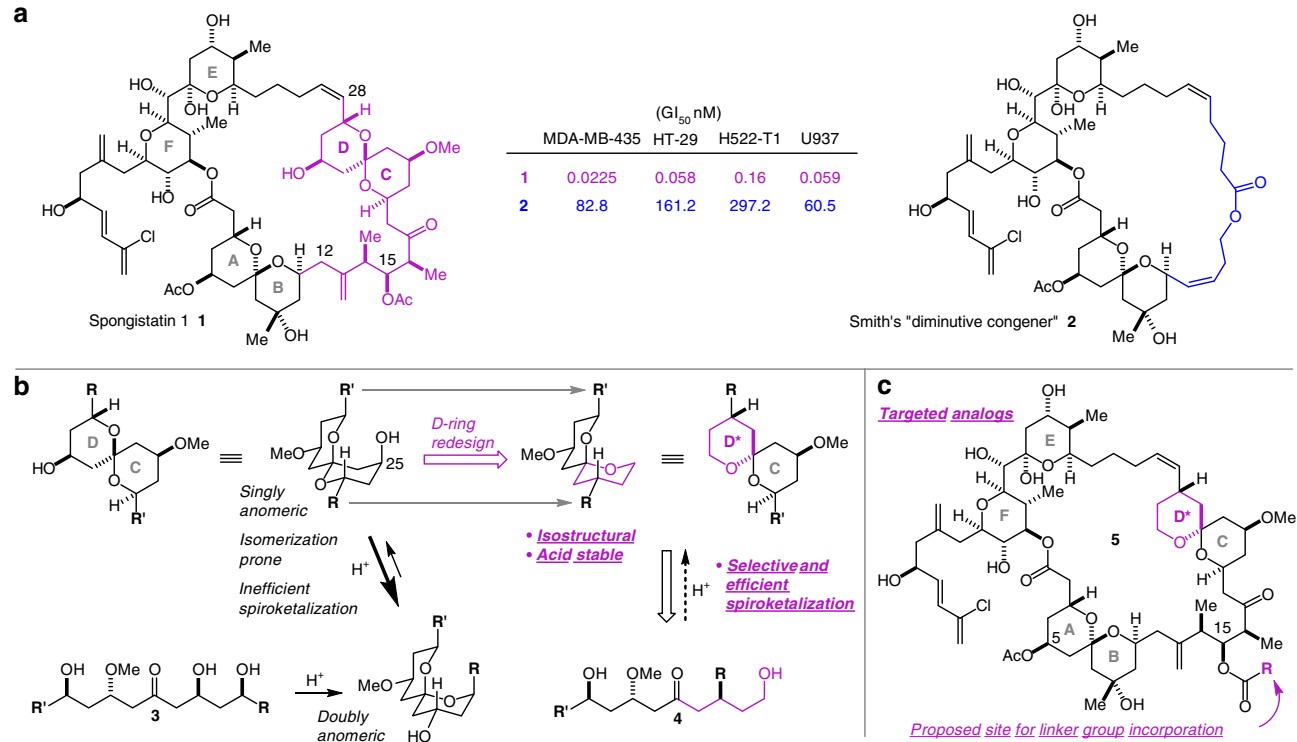

**Fig. 1** Spongistatin 1 is an extraordinarily potent anti-mitotic agent whose development is hindered by synthetic inaccessibility. **a** Smith's simplified analog **2** retains some significant potency suggesting that the C(12)–C(18) fragment and the CD spiroketal are not in contact with the receptor in the bound conformation, and that their role is to serve as scaffolding to lock the compound in its active conformation. **b** Relocation of the D-ring oxygen to the anomeric position and deletion of the C(25) alcohol is proposed to lead to an analog that is isostructural and therefore equipotent with the natural product, stable in the low pH endosomal and lysozomal environments, and significantly easier to synthesize efficiently by spiroketalization of dihydroxyketone **4**. **c** The study aims are the development of efficient synthetic access to the spongistatins, synthesis and evaluation of the D-ring modified analog **5a** (R = Me), and validation of the C(15) acyloxy group as a site for linker functional group incorporation without a significant loss of potency

**Fig. 2** Retrosynthetic analysis. The target molecules may be split into the ABCD* (**6**) and EF (**7**) fragments, which may be coupled with a Wittig reaction, followed by macrolactonization to complete the synthesis. A direct complex fragment coupling *syn*-crotylation reaction of **8** with CD* spiroketal aldehyde **9a** or **9b** was envisioned as the key step in the synthesis of the ABCD* fragment, while a direct complex fragment coupling alkallylation reaction of **10** with aldehyde **11** was envisioned as a critical step in the synthesis of the EF fragment. Crotyl- and allylsilanes **8** and **10** (M = Si) were envisioned to arise from diene **12** and allylchloride **13**, greatly simplifying the synthesis. TFA trifluoroacetate, TMSE 2-trimethylsilylethyl, TES triethylsilyl, TBDPS *tert*-butyldiphenylsilyl, TBS *tert*-butyldimethylsilyl

their reliably excellent reactivity and broad generality and to the fact that the requisite crotyl- and allylsilanes **8** and **10** (M = Si) appeared accessible from the simple precursors diene **12** (ref. [39]) and allylchloride **13** by way of mild, functional group-tolerant, and chemoselective transition metal-catalyzed hydrosilylation and allylic substitution reactions, respectively.

**Synthesis of the ABCD\* fragment.** Not knowing at the outset which of the C(17) diastereomeric aldehydes **9a** and **9b** would perform best in the critical fragment coupling crotylation

reaction, we developed a flexible and efficient route to both (Supplementary Figs. 1–3). The final 11-step route to **9b** (C(17)-*R*) is representative and entailed a 1,3-*anti* diastereoselective Mukaiyama aldol reaction[40] of aldehyde **14** with silyl enol ether **15** to give **16** in 70% yield (Fig. 3a). Methyl ether formation then delivered **17** in 83% yield, setting the stage for the vital spiroketalization reaction to form the redesigned CD* spiroketal. Treatment of **17** with pyridinium *p*-toluenesulfonate in MeOH did indeed effect a robust and highly diastereoselective spiroketalization reaction to form the desired doubly anomeric

**Fig. 3** An 18-step synthesis of the fully elaborated ABCD* fragment of the spongistatins. **a** The synthesis of the redesigned CD* spiroketals **9a** and **9b** entails as key steps a Mukaiyama aldol reaction and a diastereoselective spiroketalization reaction. **b** The direct complex fragment coupling syn-crotylation reaction to join AB spiroketal diene **12** with CD* spiroketal aldehyde **9b** to produce **25** in a single step is robust and scalable and is the crucial innovation in the development of a highly step-economical and efficient synthesis of the ABCD* fragment of the spongistatins. **c** Five straightforward steps convert **21** and **25** into the completed ABCD* fragment **6a** and linker-equipped analogs **6b** and **6c**. DBU1,8-diazabicyclo[5.4.0]-undec-7-ene, DMAP4-dimethylaminopyridine, TASF tris(dimethylamino)sulfonium difluorotrimethylsilicate

spiroketal **18** in 75% yield. Treatment of **18** with triethylsilyl triflate (TESOTf) and then trimethylsilyl triflate (TMSOTf) resulted in sequential alcohol protection and acetal deprotection to give CD* spiroketal aldehyde **9b** in 82% yield. An earlier version of this route was used to access **9a** (C(17)-S), and it was with this aldehyde that our investigation of the complex fragment coupling crotylation reaction began based on a model study which suggested that this stereochemical permutation would represent a matched case in the fragment coupling crotylation reaction[37]. Indeed, palladium-catalyzed hydrosilylation of diene **12** (ref. [41]) followed by complexation with diamine (R,R)-**19** produced crotylsilane **20**, which was employed in our first-generation Sc(OTf)$_3$-catalyzed crotylation protocol[37] with **9a** leading to the isolation of **21** with high (≥15:1) diastereoselectivity (Fig. 3b). While we were gratified by this proof-of-concept result, the requirement for isolation of the intermediate crotylsilane **20** (necessary because the DBU•HCl salts poison the Sc(OTf)$_3$ catalyst) was deemed highly problematic in that crotylsilanes such as **20** are prone to degradation when exposed to moisture both

rendering the isolation technically difficult and risky and leading to significant variability in the yield of the reaction. Motivated to address these limitations, we recently developed a second-generation diaminophenol ligand (**22**) that induces reactivity in its derived crotylsilanes ~equivalent to the diamine **19**/Sc(OTf)$_3$ combination, but without requiring the Sc(OTf)$_3$ catalysis[38], which allows its use in a direct one-pot protocol without isolation of the crotylsilane prior to reaction with the aldehyde. When this protocol was employed to generate and react (without isolation) crotylsilane **23** with **9a**, however, we were surprised to find that the reaction produced, at best, trace amounts of the desired product. Based on a computational model for the first-generation crotylsilylation reaction[42] and on Evans' comprehensive model for merged 1,2- and 1,3-stereoinduction in related aldol reactions[43], we constructed transition state model **A** for the successful reaction of first-generation crotylsilane **20**, and then applied it to the reaction of crotylsilane **23** to construct transition state model **B**. This exercise revealed that the tethering of the phenol to the silane with diaminophenol ligand **22**, which induces the greater

reactivity in the silane, also has the unintended consequence of generating a prohibitive steric interaction between the *t*-Bu group on the phenol ring and the β-OTES group on the aldehyde. We hypothesized that we could alleviate this interaction by employing diaminophenol (*R,R*)-**24** with a methyl group in place of the *t*-Bu group and by employing CD* aldehyde **9b** (C(17)-*R*), to result in a reaction that would proceed smoothly by way of transition structure **C**. Gratifyingly, this proved to be the case with the crotylsilane formed from diene **12** and (*R,R*)-**24** reacting with aldehyde **9b** to give the desired product **25** as a single diastereomer in 69% yield from diene **12** and in 81% yield from aldehyde **9b**. Crucially, because no isolation of the crotylsilane is necessary, the reaction is robust, reproducible, and scalable, and in one case was used to synthesize 1.27 g of **25** in a single run. Completion of the ABCD* fragment synthesis comprised a five-step sequence as described in Fig. 3c. This chemistry was initially worked out using **21**, and entailed, 1. acetylation of the C(15) alcohol; 2. deprotection of the TMSE ester, TFA group, and the C(17) and C(28) silyl ethers; 3. esterification of the C(1) acid with tri-*iso*-propylsilyl chloride (TIPSCl); 4. double oxidation of the C(17) and C(28) alcohols; and 5. protection of the tertiary carbinol as its TES ether to produce the completed ABCD* fragment **6a** in 50% yield over the five steps. Using this sequence starting from **25**, we also synthesized two azide-bearing ester linker candidates at C(15), **6b** and **6c**. Overall, the route to ABCD* fragments **6a–c** comprises just 18 steps in the longest linear sequence (LLS). The D-ring redesign had the intended impact on the ease and scalability of the synthesis of aldehydes **9**, while the ease and scalability of the single step conversion of **12** and **9b** to **25** made possible the rapid and far less resource- and time-intensive synthesis of the linker-bearing ABCD* fragments.

**Synthesis of the EF fragment**. Our synthesis of the chlorodiene-bearing EF fragment commenced with a double cross-metathesis reaction between diene **26** and *tert*-butyl acrylate using the second-generation Hoveyda-Grubbs (HG-II) catalyst[44] (Fig. 4). By using the acrylate as solvent we have been able to reduce the catalyst loading to an extraordinarily low level, with just 0.69 mol % catalyst resulting in isolation of **27** in 58% yield[45]. Double Sharpless asymmetric dihydroxylation (AD)[46] then delivered tetraol **28** in 65% yield as a 4.5:1 mixture of diastereomers[47]. Diastereoselective lactonization to establish the C(40) stereocenter was most effectively carried out using dry HCl in MeOH and was accompanied by transesterification to deliver lactone **29**. Without isolation, this material was subjected to triple benzyl ether protection with 2,4,6-tris(benzyloxy)-1,3,5-triazene (Tri-BOT)[48] giving lactone **30** as a single diastereomer (the minor *meso* diastereomer from the double AD reaction does not undergo the lactonization reaction) in 62% yield over two steps. Petasis methylenation[49,50] of both carbonyls gave bis enol ether **31** in 55% yield, both setting up a hydroboration/*B*-alkyl Suzuki reaction for the allyl sidechain introduction and converting the methyl ester to a methyl ketone in masked form. Chemo- and diastereoselective hydroboration of **31** with 9-borabicyclo[3.3.1] nonane (9-BBN)[51] was followed by *B*-alkyl Suzuki coupling[52] with vinyl bromide **32** and acidic hydrolysis of the methyl enol ether to deliver the fully elaborated F-ring methyl ketone **33** in 49% yield. Though we do not know the detailed mechanistic origin of the chemoselectivity of the hydroboration reaction for the cyclic enol ether, we note that attempts to accelerate the reaction by conducting it at higher temperatures led to competitive hydroboration of the acyclic methyl enol ether and lower overall yields of **33**. Introduction of the E-ring commenced with a diastereoselective aldol reaction of methyl ketone **33** with aldehyde **34** to give **35** in 84% yield with 11:1 diastereoselectivity[31,33]. Acid-promoted ketalization delivered **36** in 70% yield, and alcohol protection gave **37** in 87% yield. Four protecting group and functional group manipulations (1. reductive removal of all 5 benzyl groups[25], 2. persilylation of all 5 alcohols, 3. selective deprotection of the two primary alcohols, and 4. chlorination of both primary alcohols) delivered allylchloride **13** in 55% yield

**Fig. 4** A 16-step synthesis of the fully elaborated EF fragment of the spongistatins. Key steps include the double CM, double AD, and diastereoselective lactonization sequence that provides F-ring lactone **29** with five stereocenters in just three scalable steps, and the direct *n*-Bu₄NBr-catalyzed complex fragment coupling alkallylation reaction to incorporate the chlorodiene sidechain (**13** + **11** to **39**). (DHQD)₂PHAL hydroquinidine 1,4-phthalazinediyl diether, TMSCl trimethylsilyl chloride, PPTS pyridinium *para*-toluenesulfonate, TBSOTf *tert*-butyldimethylsilyl triflate, LiDBB lithium 4,4′-di-*tert*-butylbiphenylide

over four steps, setting up the critical complex fragment coupling alkallyation reaction. In analogy to the diene hydrosilylation/diaminophenol complexation sequence described above, activated alkallylsilanes may be accessed from the corresponding alkallyl chlorides by way of the Benkeser–Furuya reaction[53] and in situ diaminophenol complexation. Alkallysilanes derived from complex and sterically hindered alkallyl chlorides such as **13** react very slowly, however, and the attempted coupling of **38** (derived from **13** and diaminophenol **24**) with aldehyde **11** led mainly to decomposition. This impasse inspired a search for ways to accelerate these reactions and this led to the discovery that weakly coordinating anions (e.g. Br−, I−, TfO−) are effective catalysts that operate, the evidence suggests, by adding to the allylsilane and accessing small equilibrium concentrations of the more highly activated allylsilicate[54]. When the coupling of **13** and **11** was repeated in the presence of 20 mol% $n$-Bu$_4$NBr, the desired product **39** was produced as a single diastereomer (≥20:1 dr) in 71% yield (94% based on recovered allylchloride **13**), presumably through the intermediacy of the activated allylsilicate **38•Br−**. Alcohol protection was followed by a one-pot Finkelstein reaction and phosphonium salt formation[23] to deliver the fully elaborated EF fragment **7** in 83% yield over two steps. Overall, this synthesis of the EF fragment comprises just 16 steps in the longest linear sequence. This step economy derives in large part from the development of a six-step synthesis of F-ring methyl ketone **33**, and the direct fragment coupling alkallylation reaction to install the chlorodiene sidechain and C(47) stereocenter in a single step.

**Completion of the synthesis and biological evaluation**. Wittig coupling of EF fragment **7** with aldehydes **6a–c** proceeded smoothly and delivered **40a–c**, in 58–63% yields (Fig. 5a). Selective deprotection of the TIPS ester and the C(41) and C(42) TES groups was carried out with $n$-Bu$_4$NF•3H$_2$O to give the hydroxy acids **41a–c** in 79–84% yields, which were subjected to macrolactonization employing the Yamaguchi protocol[55] to give

the macrolactones **42a–c** in 77–87% yields. Global deprotection of the remaining silyl ether-protecting groups as well as E-ring ketal hydrolysis was effected with aqueous HF to deliver the target structures **5a–c** in 82–83% yields. These compounds were evaluated for in vitro cytotoxicity against the 1A9 (A2780, ovarian), PC3 (prostate), Ca46 (lymphoma), and U937 (lymphoma) cell lines, using paclitaxel as a reference compound in the absence of a sample of spongistatin 1 (Fig. 5b; Supplementary Table 1). First and most critically, the data reveal that **5a** is characterized by potencies in a similar range (i.e. low picomolar) to the parent natural product (Smith has reported a potency for spongistatin 1 of 0.059 nM against the U937 cell line[36]), consistent with our hypothesis regarding the primary role of the CD spiroketal as structural/conformational scaffolding and validating the D-ring redesign. Second, the data show that whereas the benzoate derivative **5b** is more than an order of magnitude less potent than **5a**, the aliphatic ester **5c** is only slightly (~three-fold) less potent than **5a**. While caution must be exercised when attempting to rationalize such small changes in potency, these data are consistent with the hypothesis that whereas the change from an acetate to a benzoate results in significant local conformational changes in the vicinity of C(15), the extension of an acetate to a higher order alkyl ester does not. While further optimization of the structure of the C(15)-azide-bearing acyl group to minimize any loss in potency may be desirable, these results constitute a proof-of-concept that the C(15)-acetate may be structurally modified without a significant loss of activity.

## Discussion

Some 70% of all ADCs currently in clinical development use either a maytansinoid or an auristatin as the payload[56], mainly because they are two of the only drugs that meet the criteria, including that they are readily available in significant quantity. Even within the microtubule-targeting drug class, it may be that that certain payloads perform more effectively against certain

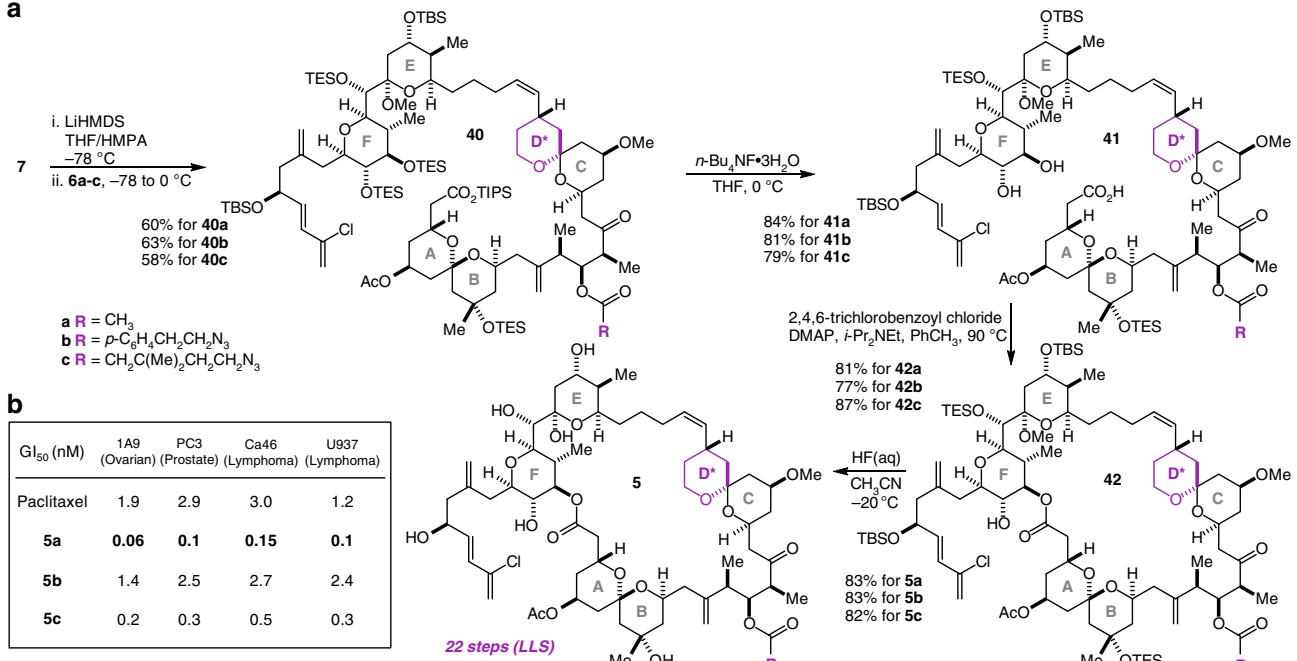

**Fig. 5** Completion of the syntheses and cell growth inhibition data. **a** Four steps couple the ABCD* and EF fragments and produce the spongistatin 1 analogs **5a–c**. **b** Cell growth inhibition assays with four human cancer cell lines establish that the D-ring modified analog **5a** is approximately equipotent with spongistatin 1, while **5c** represents a proof-of-concept result that the C(15) acetate may be structurally modified with only a minimal reduction in potency. LiHMDS lithium hexamethyldisilazide, HMPA hexamethylphosphoramide

| GI$_{50}$ (nM) | 1A9 (Ovarian) | PC3 (Prostate) | Ca46 (Lymphoma) | U937 (Lymphoma) |
|---|---|---|---|---|
| Paclitaxel | 1.9 | 2.9 | 3.0 | 1.2 |
| **5a** | **0.06** | **0.1** | **0.15** | **0.1** |
| **5b** | 1.4 | 2.5 | 2.7 | 2.4 |
| **5c** | 0.2 | 0.3 | 0.5 | 0.3 |

cancers than others[57], and an expanded pool of potential payloads for ADC development could advance the field in significant ways[58]. Many of these drugs are structurally complex natural products that are unavailable in quantity from natural sources and/or that do not have a synthetically selective and pharmacologically innocent linker functional group, however, and while ideally the limits of chemical synthesis would never prevent efforts to develop such compounds, the reality is that, especially with compounds as complex as the spongistatins, lengthy syntheses are often required that do not readily lend themselves to these kinds of drug development efforts. Indeed, the previous syntheses of spongistatin 1 and 2 have all required ≥30 steps in the longest linear sequence and ≥105 total steps. Due to the development of powerful and uniquely enabling new methodologies and strategies including the direct diaminophenol-activated complex fragment coupling crotylsilylation and $n$-Bu$_4$NBr-catalyzed alkallylsilylation reactions, the synthesis described here proceeds in 22 steps in the longest linear sequence (1.4% overall yield) and 68 total steps. This is important not only in advancing the frontiers of step economy and efficiency in the synthesis of non-aromatic polyketide natural products[59–62], but also in facilitating the compound development efforts necessary to begin evaluating whether the spongistatins, properly configured for use in ADCs and other conjugates, can serve as effective payloads. In the present case, this step economy and efficiency have allowed us to validate the D-ring redesign and rendered the synthesis and evaluation of linker functional group-equipped analogs **5b** and **5c** a far less laborious and time- and resource-intensive effort and have allowed us to secure a crucial proof-of-concept result in the form of **5c**, a highly potent linker-equipped analog of the spongistatins. Having validated the D-ring redesign and the C(15) acyl group linker strategy, our ongoing efforts are focused on a campaign to produce significant quantities of both **25** and **7** in order both to further optimize the structure of the C(15) acyl linker group for full potency retention and to use the optimized structure in the preparation and evaluation of conjugates.

## Methods

**General**. All reactions were carried out under an atmosphere of nitrogen in flame-dried glassware with magnetic stirring unless otherwise indicated. Degassed solvents were purified by passage through an activated alumina column. Thin-layer chromatography (TLC) was carried out on glass backed silica gel TLC plates (250 µm) from Silicycle; visualization by UV light, cerium ammonium molybdate (CAM), phosphomolybdic acid (PMA), $p$-anisaldehyde stain, or potassium permanganate (KMnO$_4$) stain. HPLC analysis was carried out on an Agilent 1200 Series using a Chiralpak OD-H (250×4.5 mm ID) column. Diastereomeric ratios for all compounds were determined by $^1$H NMR analysis of the unpurified reaction mixtures. $^1$H NMR spectra were recorded on a Bruker AVIII 400 (400 MHz) and AVIII 500 (500 MHz) spectrometer and are reported in ppm, relative to residual protonated solvent peak (CDCl$_3$, 7.26 ppm) unless otherwise indicated. Data are reported as follows: bs = broad singlet, s = singlet, d = doublet, t = triplet, q = quartet, m = multiplet, dd = doublet of doublets, ddd = doublet of doublet of doublets, ddt = doublet of doublet of triplets, td = triplet of doublets; coupling constant(s) in Hz; integration). Proton decoupled $^{13}$C NMR spectra were recorded on a Bruker AVIII 400 (100 MHz) and AVIII 500 (125 MHz) spectrometer and are reported in ppm from CDCl$_3$ internal standard (77.00 ppm) unless otherwise indicated. High-resolution mass spectra were obtained from the Columbia University Mass Spectrometry Facility on a Waters XEVO G2XS QToF mass spectrometer equipped with a UPC2 SFC inlet, electrospray ionization probe, and atmospheric pressure chemical ionization probe. Infrared spectra were recorded on a Perkin Elmer Paragon 1000 FT-IR spectrometer. Optical rotations were recorded on a Jasco DIP-1000 digital polarimeter. pH 7.00 buffered silica gel was prepared as follows: A 1000 mL round bottom flask charged with silica gel (250 g) and 25.0 mL of pH 7.00 buffer solution (potassium dihydrogen phosphate/sodium hydroxide; Fluka Analytical) was stirred on rotavapor under atmospheric pressure overnight.

## Data availability

The authors declare that the data supporting the findings of this study are available within the paper and the Supplementary Information as well as from the authors upon reasonable request.

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

## Acknowledgements

Financial support from the National Institute of General Medical Sciences (R01GM058133 to J.L.L., and postdoctoral fellowship (F32GM090487) to P.S.T.), the National Science Foundation (Graduate Research Fellowships to L.M.S. and M.A.T.-S.), the American Cancer Society (postdoctoral fellowship (124489-PF-13-311-01-CDD) to K.S.W.), and the Intramural Research Program of the Eunice Kennedy Shriver National Institute of Child Health and Human Development, National Institutes of Health is gratefully acknowledged.

## Author contributions

J.L.L. designed and directed the research project and wrote the manuscript. D.L.S. performed the assays described in Fig. 5b and Supplementary Table 1, and made contributions to the project design. Building on critical foundational work by S.K.R. and P.S. T., L.M.S., M.A.T.-S., K.S.W., and J.R.I. all made major contributions to the work reported here. T.D.C. made contributions to the synthesis of 9b described in Fig. 3a and Supplementary Fig. 1.

## Additional information

**Competing interests:** The authors declare no competing interests.

