## [Peer Review File · Nature Communications]

Reviewer #1 (Remarks to the Author):

The authors report a brilliant study on a structural modification of spongistatin 1 that replaces a singly anomeric and acid-labile spiroketal with a doubly anomeric and thermodynamically-favored motif. Since the rearrangement cannot accommodate the analogous D-ring hydroxyl, this group is deleted.

I cannot emphasize enough the importance of this manuscript; it is strongly recommended for acceptance. This sort of approach to precision modification and structural stabilization of natural products has general predecessors in the field, especially via Boger (JOC, 2017, 11961). But Leighton provides an important foundation for polyketides, especially in the interplay of structural stability, synthetic design and biological activity. The authors may wish to cite a recent example in the terpenoids (ACS Central Science, 2017, 1329), not just because of its conceptual similarity, but to build momentum in the field and bolster enthusiasm for this approach. This paper could be a historical example that signals a sea change in how natural products total synthesis is carried out.

The manuscript describes the design and synthesis of an equipotent yet synthetically simpler spongistatin 1 analog derived from extensive analysis of previous syntheses and SAR studies of the spongistatins. The author's recognized the synthetic difficulty attributed to the thermodynamically less stable, singly anomeric, CD spiroketal and postulated that modification of this ring to form an isostructural, and more acid stable spiroketal would drastically simplify the synthesis without affecting the biological properties. Acylation of the C15 hydroxyl group with azide containing side chains could potentially lead to Antibody Drug Conjugate (ADC) analogs. The linear precursor to the structurally modified CD spiroketal was rapidly assembled and selectively cyclized into the desired spiroketal under acidic conditions without the need for any re-equilibration to form the desired product.

The authors then utilized their modified crotylation conditions to reproducibly, in a scalable manner, combine the AB and CD subunits after tuning the diaminophenol ligand to minimize undesired steric interactions that were shutting down the reaction. The authors have also acylated the C15 hydroxyl group with different azide-containing linkers to append antibodies or nanobodies. The EF fragment was coupled to the diene fragment through another crotylation that was facilitated by weakly coordinating anions. Can the authors comment on whether Br⁻ activation increases Lewis acidity of the silane through a two-step aldehyde complexation/ carbonyl addition? Or whether the silicate undergoes a concerted, bimolecular reaction with aldehyde?

The synthesis was completed through a similar sequence to the previous syntheses of the spongistatins. However, the synthesis is much shorter in both LLS and total steps which is crucial for

analog development and biological testing. Critically, the spongistatin analog that contains only the modified CD ring has comparable cytotoxicity to spongistatin 1 (at least, when standardized against taxol, which is acceptable).

Interestingly, the azide containing benzoate is substantially less active, while the aliphatic azide only loses three fold potency. This is important in that it proves that these unnatural analogs could prove to be viable ADC payloads.

To reiterate, I strongly recommend the manuscript for acceptance. The authors may wish to differentiate this approach from Wender's, which reduces structural complexity to simplify synthesis. The authors have not really changed structural complexity, yet rendered the synthesis simpler by virtue of their structural perturbation. Similarly, Smith's work significantly displaces spongistatin from natural product space and loses much of its benefits. The authors have achieved something unique here and I hope to see more in this vein. With any luck, we may see a drug and a reinvigoration of natural products research.

Reviewer #2 (Remarks to the Author):

This manuscript describes the design and synthesis of spongistatin 1 analogues toward the development of an ADC payload for cancer treatment. The features of this work are: 1) redesign of the CD-ring spiroketal for improved compound stability; 2) development of fragment-assembling allylation/crotylation; and 3) identification of the C15 acetate as a suitable modification point for ADC development.

Marine polyketide macrolides are expected to be an important source of new cancer chemotherapeutics. However, their scarce availability from natural sources as well as their structural complexity that necessitates extensive synthetic efforts are serious problems toward that end. Leighton and co-workers have successfully addressed these problems by targeting rationally designed functional analogues and by inventing new synthetic methods that streamline the overall synthesis. Although the authors only provided in vitro cytotoxicity data, it is likely that the analogues 5a and 5c are more promising lead compounds than the natural product itself. Importantly, this work not only advanced the chemistry of spongistatins but also has a conceptual impact on the development of new therapeutics based on complex polyketide natural products. I support publication of this work in Nature Communications after consideration of the following minor points:

1. The authors claim that the synthesis summarized in this manuscript is highly step-economical. I think it would be much easier to recognize that point by including in the main text how many steps were required for the preparation of 12 and 14 (or 15).
2. The authors have successfully transformed bis-enol ether 31 to methyl ketone 33 by means of a chemoselective hydroboration of the exocyclic enol ether moiety. Why the methyl enol ether moiety remained intact under these conditions?
3. According to SI, the authors have prepared so far 15 mg of 5a and 4 mg of 5c. Considering the potency of the parent natural product, the amount of 5a/5c prepared would be sufficient for in vitro or cell-based assays. However, are they enough for future ADC development study?

Point-by-point responses to the referees' comments:

Reviewer #1. Reviewer 1 was highly positive about the paper, and we sincerely thank the referee for their very kind comments on the importance and quality of this work.

The referee noted that, “This sort of approach to precision modification and structural stabilization of natural products has general predecessors in the field” and provided two references (Boger, JOC; Shenvi, ACS Central Science). We agree that these are relevant and important references to include (and regret and apologize for their omission in the original version of the manuscript), *and have included them as new references 32 and 33.*

The referee asked, “Can the authors comment on whether the Br⁻ activation increases Lewis acidity of the silane through a two-step aldehyde complexation/carbonyl addition? Or whether the silicate undergoes a concerted, bimolecular reaction with aldehyde?” Everything we know about the mechanism was published in our 2017 Org. Lett. Paper, where we provided compelling evidence that the bromide catalyst does add to the silane to form a silicate that subsequently reacts with the aldehyde. If I understand correctly, the reviewer is asking a more subtle question about whether the reaction of the aldehyde with the silicate is a single step process or a two-step process that involves the aldehyde-bound silicate as an intermediate. The short answer is that we do not have any evidence to determine this either way, and we cannot comment usefully on this. Alternatively, if the reviewer is asking about a mechanism wherein there is no interaction between the aldehyde oxygen and the silicon, we think this is easily ruled out by the fact that these reactions give diastereo- and enantioselectivities that are similar to the uncatalyzed reactions and consistent with the usual Type I closed transition state and inconsistent with an open transition state of the type that such a mechanism would imply. Regardless, we are very interested in the answer to this and many other related questions about this mode of catalysis as we continue to develop these reactions, and any mechanistic elucidation that we achieve in the course of these ongoing studies will be published as it becomes available.

The referee noted that our assay results are, “standardized against taxol, which is acceptable.” We appreciate the reviewer’s comment, and note that since submission of the manuscript, our collaborator Dr. Dan Sackett has performed additional assays with the U937 (lymphoma) cell line. Smith has reported a GI₅₀ value for spongistatin 1 of 0.059 nM against the U937 cell line, whereas Dr. Sackett finds a GI₅₀ value for our analog **5a** of 0.1 nM. Though this is still not the ideal side-by-side direct comparison of spongistatin 1 vs. **5a**, it nevertheless does provide additional evidence that **5a** is essentially equipotent with spongistatin 1 to a first approximation, and that if there are statistically significant differences in potency, they are likely very small. *We have added these new assay results to Fig. 5b and Supplementary Table 1, and we have added a brief statement about this new assay in the text that accompanies Fig. 5b.*

Finally, the reviewer suggested that we “may wish to differentiate this approach from Wender’s, which reduces structural complexity to simplify synthesis. The authors have not really changed structural complexity, yet rendered the synthesis simpler by virtue of their structural perturbation.” We again thank the referee for their kind words and their very careful reading and

consideration of some of the broader themes of this work. We agree with the reviewer on the concept, but note that this is a particularly subtle distinction that in our opinion would be difficult to address in a couple of sentences and that may be better discussed in a perspective or review-type article. In the present context, it is our opinion that getting into such subtleties in this manuscript would risk rendering the discussion in the introduction too broad and taking focus away from the main points of the paper.

Reviewer #2. Reviewer 2 provided a positive review of the work, and we thank the referee for their positive comments and outlook on the work. The reviewer recommended publication “after consideration of the following minor points:”

1. “The authors claim that the synthesis summarized in this manuscript is highly step-economical. I think it would be much easier to recognize that point by including in the main text how many steps were required for the preparation of 12 and 14 (or 15).”

*In response to this comment we have changed the sentence “The final route to **9b**...” to “The final 11-step route to **9b**...”*

2. “The authors have successfully transformed the bis-enol ether 31 to methyl ketone 33 by means of a chemoselective hydroboration of the exocyclic enol ether moiety. Why the methyl enol ether remained intact under these conditions?”

We do not have a simple explanation for this, but we note that if the reaction is carried out at higher temperatures we do indeed start to see hydroboration of the methyl enol ether as well. When we first decided to try this reaction, we did not know if we could obtain useful levels of chemoselectivity, and were delighted to find that we could. But again, we do not have any data that would help us develop a detailed mechanistic explanation for the observed chemoselectivity.

3. “According to SI, the authors have prepared so far 15 mg of 5a and 4 mg of 5c. Considering the potency of the parent natural product, the amount of 5a/5c prepared would be sufficient for in vitro or cell-based assays. However, are they enough for future ADC development study?”

The answer to this question is no, 4 mg of 5c is not enough material to do any substantial ADC development study. This is clearly the direction we are going with this project, but we hope that the reviewer can appreciate that before we could initiate such development efforts, we first had to develop more efficient synthetic access (what this paper is primarily about), and then establish that the C(15) acyloxy groups are the correct strategy for linker incorporation. It would have been risky and even foolish to devote significant effort to scaling up the synthesis *before* we had validated the D-ring redesign and the C(15) linker incorporation. Now that we have done so, we are gearing up to do a scale-up of the synthesis so that we can move the project into this next phase.